Estimating density of native carnivores in central Chile landscapes using a simulated movement model, cameratrapR: insights on their potential exotic prey dietary subsidy

Ramírez-Alvarez Diego diego.ramirez@sag.gob.cl 1
Li Xinhai lixh@ioz.ac.cn xinhai_li_edu@126.com 2 3
1 Unidad de Vida Silvestre, Servicio Agrícola y Ganadero, Región de O’Higgins , Rancagua , Chile
2 Key Laboratory of Animal Ecology and Conservation Biology, Institute of Zoology, Chinese Academy of Sciences , Beijing , China
3 University of Chinese Academy of Sciences , Beijing , China
Hedrick Ann
Electronic publication date: 2025 Sep 1
Publication date: 2025
Volume: 13
Electronic Location ID: e19946
Received 2024 Sep 30; Accepted 2025 Jul 28
Copyright: ©2025 Ramírez-Alvarez and Li
Copyright year: 2025
Copyright holder: Ramírez-Alvarez and Li
License: This is an open access article distributed under the terms of the Creative Commons Attribution License, which permits unrestricted use, distribution, reproduction and adaptation in any medium and for any purpose provided that it is properly attributed. For attribution, the original author(s), title, publication source (PeerJ) and either DOI or URL of the article must be cited.
License URL: https://creativecommons.org/licenses/by/4.0/

Keywords: CameratrapR, Dietary subsidy, Ecological subsidy, Non-native prey, Simulated movement model, Camera trapping, Carnivores, Population density

Funding: Universidad de O’Higgins and Codelco El Teniente Chile, the Intergovernmental International Science and Technology Innovation Cooperation Program Under National Key Research and Development Plan 2024YFE0198600 Third Xinjiang Scientific Expedition Project 2021XJKK1302 This research was funded through a collaboration agreement between Universidad de O’Higgins and Codelco El Teniente Chile, the Intergovernmental International Science and Technology Innovation Cooperation Program Under National Key Research and Development Plan (2024YFE0198600), and the Third Xinjiang Scientific Expedition Project (Grant No. 2021XJKK1302). There was no additional external funding received for this study. The funders had no role in study design, data collection and analysis, decision to publish, or preparation of the manuscript.

==============================
Background

Species-specific density is an essential parameter for evaluating population stability and ecosystem health. We estimate the population density of native carnivores, South American gray and culpeo foxes (Lycalopex spp.), guiña (Leopardus guigna) and Andes skunk (Conepatus chinga), in central Chile, inferring on the potential influence of the availability of introduced exotic prey on their population dynamics.

Methods

Monitoring with camera traps was conducted from March 1, 2021, to March 31, 2022, across three distinct landscapes representative of the coastal mountain range of central Chile: Mediterranean coastal thorn forest, Mediterranean coastal sclerophyllous forest, and exotic monoculture tree plantations. We estimated density using an R package, cameratrapR, where animal movement was simulated using a correlated random walk within the camera grids. Simulations were run for a range of numbers of individuals, representing a gradient of population densities. We matched these results and the observed photo records using a machine learning algorithm, random forest, thereby estimating population density and its 95% confidence intervals.

Results

A total of 10,046 trap days resulted in 9,120 photographs of carnivores, with 3,888 independent records. Our findings indicate that Lycalopex spp. are the carnivore species with the highest population density in central Chile (4.1–4.8 individuals/km2). Furthermore, their density was higher in the exotic monoculture tree plantation ecotype compared to native forests, suggesting a dietary subsidy from non-native prey. We report the first population density estimates for Conepatus chinga (1.8–2.2 individuals/km2) and Leopardus guigna (0.7–1.3 individuals/km2) in the three landscapes, showing different habitat preferences and resource availabilities across landscapes. The results highlight the complex interactions between carnivores and their environments, particularly the role of exotic introduced species as prey items in shaping native carnivore populations. These findings also provide insights into carnivore adaptability and the implications for biodiversity conservation.

Introduction

Carnivores play an essential role in the functioning of natural ecosystems, in some cases acting as umbrella species generally requiring large habitat areas to maintain viable populations, which benefits many other species and natural communities with smaller habitat areas (Noss et al., 1996; Thorne, Cameron & Quinn, 2006; Garcia et al., 2020). Populations of different carnivores interact in various ways, and conservation efforts require a greater understanding of the complex relationships between species at the landscape level (Glen & Dickman, 2005).

Landscapes can be selectively used by certain carnivore species, leading to habitat specialization, which may limit the establishment of a population according to the fulfilment of its specific requirements (Mitchell et al., 2012). Conversely, habitat generalists are adaptable in using various landscapes, making these habitats less of a limiting resource (Boitani & Powell, 2012). New methodologies in carnivore occupancy modeling provide relevant data to understand the multiscale nature of habitat use patterns (Lewis et al., 2015; Chanchani et al., 2016). The response of carnivores to habitat heterogeneity is shaped by intrinsic factors such as movement capacity and individual traits, including size, sex, and age; and extrinsic factors such as prey availability, landscape structure, interspecific competition and anthropogenic pressure (Grassel, Rachlow & Williams, 2015; Moreira-Arce et al., 2021). Furthermore, advancements in methodology have introduced tools specifically designed to analyze animal movement patterns, such as the R package aniMotum (Jonsen et al., 2023), enhancing our ability to study these dynamics (Jonsen et al., 2023).

The Central Chile Coast Range extends approximately 3,000 km north-south, separating the intermediate depression from the coastal plains (Cunill, 1979). Geologically older than the Andes Range, it is a descending formation with its highest summit reaching 3,114 m (Suarez, Naranjo & Puig, 2010). There are three different landscapes or vegetation macrohabitats in central Chile Coast Range: (a) Mediterranean coastal sclerophyllous forest (SF) (composed mainly of Lithrea caustica, Quillaja saponaria, Cryptocarya alba, Peumus boldus and Azara integrifolia), (b) Mediterranean coastal thorn forest (TF) (composed mainly of Acacia caven, Maytenus boaria, Trevoa trinervis and Talguenea quinquinervia) and (c) Exotic monoculture tree plantations (MP) (Pinus radiata and Eucalyptus globulus), (Luebert & Pliscoff, 2018). The first two native landscape areas have shrunk and are highly fragmented due to their conversion to introduced exotic monoculture tree plantations, which have gradually increased their area since the 1970s, negatively impacting the native biodiversity (Heilmayr et al., 2016; Astorga-Schneider & Burschel, 2019; Becerra & Simonetti-Zambelli, 2020).

Seven carnivores species have been described occupying the central Chile Coast Range: South American gray fox (Lycalopex griseus), culpeo fox (Lycalopex culpaeus), puma (Puma concolor), guiña (Leopardus guigna), pampas cat (Leopardus colocola), Andes skunk (Conepatus chinga) and lesser grison (Galictis cuja) (Ramírez-Alvarez, 2018; Ramírez-Álvarez et al., 2023).

Lycalopex griseus and Lycalopex culpaeus (IUCN conservation status: least concern) are distributed along the Chilean territory, both described as habitat generalists (Salvatori et al., 1999; Zuñiga, Muñoz Pedreros & Fierro, 2009). Puma concolor (least concern) has scarce but healthy populations in the Coast Range of central Chile, also described as habitat generalist (Ramírez-Álvarez, Napolitano & Salgado, 2021; Ramírez-Álvarez et al., 2023). Leopardus guigna (vulnerable) is distributed latitudinally in Chile between the Coquimbo (S29°02′) and Aysén (S49°16′) regions, with two subspecies, Leopardus g. tigrillo (northern subspecies, assessed in this study) and Leopardus g. guigna (southern subspecies) (Napolitano et al., 2014; Napolitano et al., 2015b; Napolitano et al., 2020). They are closely associated with native Mediterranean forests and temperate rainforests (Dunstone et al., 2002; Sanderson, Sunquist & Iriarte, 2002; Schüttler et al., 2017), but able to adapt to human-altered and fragmented landscapes, moving across the human-dominated agricultural matrix (Gálvez et al., 2013; Galvez et al., 2021; Napolitano et al., 2015a; Napolitano et al., 2015b; Ramírez-Álvarez et al., 2023), and even using monoculture pine plantations (Acosta-Jamett & Simonetti, 2004; Ramírez-Álvarez et al., 2023). The distribution of Leopardus colocola (near threatened) across South America has been described with seven proposed subspecies (Kitchener et al., 2017) or species (Nascimento, Cheng & Feijó, 2021), with genetic and ecological niche differences along its range. It is distributed along the entire Chilean territory, using diverse habitat, but it is scarce and hard to record in central Chile. C. chinga and G. cuja (least concern) are scarcely studied in Chile but both of them have a wide distribution in South America, occupying a wide diversity of environments, being rather habitat generalists, but with a suggested niche segregation between them (Donadio et al., 2001; Tellaeche et al., 2014).

Spatial dynamics of carnivores are linked to local-scale resource availability (Walton et al., 2017), suggesting an inverse association between home range and local productivity (Zúñiga et al., 2022). Low food availability forces greater movements in search of food (Šálek, Drahnikova & Tkadlec, 2015).

Morphological differences between Lycalopex griseus and Lycalopex culpaeus are well described (Ramírez-Alvarez, 2018); however, some individuals have ambiguous characteristics, which make rapid visual identification of the species difficult in the field. In recent years some authors have even proposed the existence of hybridization, introgression, or incomplete lineage sorting within the genus Lycalopex (Silva, 2015; Tchaicka et al., 2016; Chemisquy et al., 2019; Moreira-Arce et al., 2020; Pizarro et al., 2024). The use of camera traps to identify Lycalopex species can generate errors in animals of similar morphology and/or under certain adverse visual environmental conditions (Yoshizaki et al., 2009; Foster & Harmsen, 2012; Johansson et al., 2020).

When carnivores live in sympatry, some dimensions of their ecological niche can overlap, but coexistence is possible due to segregation strategies, with differential use of space and time being among the most common (Gil-Sánchez et al., 2021; Arias-Alzate et al., 2022). Landscape characteristics and available habitats shape carnivore community composition according to resource availability, modifying ecological parameters such as species richness and population density (Andradetic monoculture tree plantation-Núñez & Aide, 2010), even in human-dominated exotic tree plantation landscapes (Lyra-Jorge et al., 2010; Lantschner, Rusch & Hayes, 2012).

Population density, defined as the number of individuals of a species in a specific area at a given time, provides essential information about population stability, dynamics within the community, environmental preferences, and responses to conservation strategies (Begon, Townsend & Harper, 2006; Primack, 2012). In carnivores, population density can be influenced by habitat quality and interspecies competition, which are crucial factors in wildlife management (Fernández & Mateos, 2017).

In this study, using camera traps, we detected native carnivore species in three different types of landscapes in central Chile, and applied a simulated movement model using the R package, cameratrapR (Li et al., 2022), to estimate their density by species at each landscape type. Based on these results, we discuss the role of introduced exotic prey as a potential non-native dietary subsidy shaping the population structure of native carnivores in central Chile. We hypothesize that carnivore species in central Chile will be well adapted to different types of landscapes, some of which are heavily anthropogenically modified, where they primarily rely on invasive exotic prey. Additionally, we obtained the first density estimates for some of these species in central Chile, an essential parameter for implementing effective conservation strategies.

Materials & Methods

Study area

Using the same study design as Ramírez-Álvarez et al. (2023), we studied three different landscapes or vegetation macrohabitats (the composition of which was described in the introduction section) in the Coast Range of the O’Higgins region, central Chile: (A) La Estrella site, composed of native Mediterranean coastal thorn forest (TF); (B) Alto Colorado site, composed of exotic monoculture tree plantations (MP); and (C) Callihue site, composed of native Mediterranean coastal sclerophyllous forest (SF) (Fig. 1). In each of these landscapes we selected an area of 1,000 ha with a homogeneous composition of its representative landscape. In each area, considering accessibility, we distributed ten camera traps, separated by 1,000 m each, in a uniform monitoring grid (O’Connell, Nichols & Karanth, 2010; Trolliet et al., 2014) (Fig. 1).

Figure 1 Three different sampled landscapes in the Coast Range of the O’Higgins region in central Chile.

Upper row: Light blue areas represent Mediterranean coastal thorn forest (TF), where we studied site (A), La Estrella; purple areas represent exotic monoculture tree plantations (MP), where we studied site (B), Alto Colorado; and green areas represent Mediterranean coastal sclerophyllous forest (SF), were we studied site (C) Callihue. Gray background represents urbanized or agricultural areas. Middle row: Pictures of the three studied landscapes. Bottom row: Red dots show camera trap distribution in the three sites.

Data collection

Photographs were collected as previously described in Ramírez-Álvarez et al. (2023). Specifically, in each installation point, at a height of approximately 40 cm, we oriented each camera, horizontally to the ground, towards trails, passageways or other areas with attributes indicating a potential fauna passage. No bait or lure was used. Based on our own previous tests, we estimate a probable detection radius of 20 m at the installation point, with image capture occurring at a 45° angle in front of the camera. We used Bushnell 24MP Trophy Cam, model 119719CW, with the following setting: Mode: Camera, Image size: HD pixel, Capture number: 2 photos, Interval: 3 S, Sensor level: Auto, Camera mode: 24 hrs. The cameras were active between March 1, 2021, and March 31, 2022, with a total photo-trapping effort of 10,046 camera days (3,098 in TF, 3,446 in MP, 3,502 in SF). We checked cameras in the field every two months, replacing the batteries. Photos were inspected visually, and only those recording native carnivores were selected for further analysis.

We used the CameraSweet programs developed by Sanderson & Harris (2013) for classification, organization and analysis of camera trap data, freely downloaded from the Small Wild Cat Conservation Foundation website (Small Wild Cat Conservation Foundation (SWCCF), 2022). We ran the SpecialRenamer, DataOrganize and DataAnalyze programs following the methodology described by Vásquez-Ibarra, Cortés & Silva-Rodríguez (2021). To ensure stable estimates as indicated by previous studies with camera traps, we defined independent events as those photos separated by >60 min (Beaudrot et al., 2016; Bahaa-el din et al., 2016; Henrich et al., 2022). To avoid mistakes in the species identification and classification of the native foxes Lycalopex culpaeus and Lycalopex griseus, since they are nearly indistinguishable by visual analysis of some photographs (mainly at night), and based on their relatively similar ecologies, we grouped the two species functionally as Lycalopex spp. canids.

Estimating population density

We estimated animal density using an R package, cameratrapR (Li et al., 2022), which can be installed from a GitHub repository at https://github.com/Xinhai-Li/cameratrapR (DOI: 10.5281/zenodo.14335558). In the package, animal movement was simulated using a correlated random walk within the camera grids, following species-specific movement parameters such as footprint chain tortuosity and homerange sizes. The simulations were run for a range of numbers of individuals (e.g., numbers from 1 to 20), representing a gradient of population densities. At last, we matched the simulated results and the observed photo records using a machine learning algorithm, random forest, to estimate the population density and its 95% confidence interval (Li et al., 2022).

For the species-specific movement parameters, such as step length (a parameter for movement simulation, not real step length), deflection angle between steps, and monthly activity levels (number of steps), we did not conduct field surveys to obtain new data. Instead, we utilized previously measured parameters from our earlier studies on several carnivore species, including the red fox (Vulpes vulpes), Corsac fox (Vulpes corsac), raccoon dog (Nyctereutes procyonoides), Eurasian lynx (Lynx lynx), leopard cat (Prionailurus bengalensis), Yellow-throated marten (Martes flavigula), sable (Martes zibellina), and Siberian weasel (Mustela sibirica) (Table S1) (Li et al., 2022). For the South American gray and culpeo foxes, we used the mean parameter values from the red fox, Corsac fox, and raccoon dog. These species share a similar lifestyle—being highly active runners—and have comparable body weights: gray and culpeo foxes (3.1–6.3 kg), red fox (5–7 kg), Corsac fox (4–5.5 kg), and raccoon dog (3–6 kg). The simulation parameters are: step length (13.18 m), standard deviation of step length (17.31 m), standard deviation of angular deflection (26.31°), number of steps in a month (5,000). For the guiña, a small wild cat, we used the mean parameter values from the Eurasian lynx and leopard cat. The guiña’s body weight falls between that of the two cats: guiña (2–3.2 kg), Eurasian lynx (11.7–29 kg), and leopard cat (1.5–5 kg). In terms of movement patterns, body weight does not play a major role. The movement parameters for the guiña are: step length (19.03 m), standard deviation of step length (13.07 m), standard deviation of angular deflection (41.04°), number of steps in a month (3,000). For the Andes skunk, we used the mean parameter values from the yellow-throated marten, sable, and Siberian weasel. These species share a similar lifestyle, and their body weights are comparable: Andes skunk (2.3–4.5 kg), yellow-throated marten (2–3 kg), sable (0.8–1.8 kg), and Siberian weasel (0.2–0.5 kg). The parameters are: step length (12.77 m), standard deviation of step length (15.16 m), standard deviation of angular deflection (52.12°), number of steps in a month (4,000). We assume that the movement patterns of carnivores in Central Chile are comparable to those observed in China, and therefore, we applied the average values of the movement parameters from corresponding species.

For home range sizes, we use the average values for each species described in the literature: For Lycalopex culpaeus: 3.5 km2 (Johnson & Franklin, 1994), 4.65 km2 (Salvatori et al., 1999), six km2 (Guntiñas et al., 2021), 6.5 km2 (Castellanos et al., 2021), 9.2 km2 (Novaro, 1997). For Lycalopex griseus: 2 km2 (Wilson & Mittermeier, 2009; Moreira-Arce et al., 2015). For Leopardus guigna: 1.25 km2 (Dunstone et al., 2002), 1.5 km2 (Sunquist & Sunquist, 2002), 11.8 km2 (Sanderson, Sunquist & Iriarte, 2002). For G. cuja: 4.7 km2 (Luengos-Vidal et al., 2016). For C. chinga: 1.09 km2 (Reppucci et al., 2009), 1.63 km2 (Kasper et al., 2012), 1.67 km2 (Castillo et al., 2011), 1.78 km2 (Castillo et al., 2013), 1.9 km2 (Donadio et al., 2001; Kasper et al., 2009).

The Random Forest model is a widely used algorithm that has been applied across numerous fields. One of its key advantages is its user-friendliness (Li, 2013); the default parameters often yield satisfactory results, unlike other models such as bagging, boosting, support vector machines, or artificial neural networks, which typically require careful calibration of parameters for optimal performance (Li & Wang, 2013). In the context of our study, the critical parameters for the Random Forest model include the number of trees used for voting and the number of variables considered at each split. We opted to increase the number of trees from the default 500 to 5,000, which we found to be sufficiently robust for our analysis. For the number of variables at each split, we retained the default setting, which is the square root of the total number of variables—in this case, the number of cameras at each landscape.

To estimate wildlife density, it is essential to follow established survey protocols (Li et al., 2024). Additionally, a certain number of images captured by camera traps is necessary for reliable population density estimation. Generally, several hundred independent records (with a minimum interval of 60 min between consecutive records) are sufficient. If the estimated density, based on hundreds of trees in a random forest model, follows a bell-shaped distribution and the 95% confidence intervals are entirely above zero, the estimate can be considered reliable. However, when the sample size is small, the estimated density may result in a flatter distribution, and the lower bound of the confidence interval may drop below zero, indicating reduced reliability.

Ethical considerations for camera trap studies

The camera traps used in this study were installed in wild areas with low probabilities of human traffic, with permission from the landowners. No humans were captured in this camera trapping, nor was any image disseminated by any means. Therefore, there was no impact on the privacy of individuals.

Results

We obtained a total of 13,112 photographs in all locations. Out of these, 9,120 images depicted carnivore species, and from these, 3,888 correspond to independents records or events (separated by at least 60 min). From these records, five species of carnivores were identified: 434 photographs of C. chinga, 38 photographs of G. cuja, 276 photographs of Leopardus guigna and 3,140 photographs of Lycalopex spp. (both species: Lycalopex griseus and Lycalopex culpaeus) (Figs. 2 & 3, Figs. S1–S2). In 34 of these photographs, we observed Lycalopex spp. taking lagomorphs as prey items.

Figure 2 The results of camera trapping at the three sites for Lycalopex griseus & Lycalopex culpaeus.

The circle sizes indicate the number of effective pictures.

Figure 3 The results of camera trapping at Alto Colorado site as example (monoculture tree plantations MP) for native carnivore species detected.

The circle sizes indicate the number of effective pictures.

We estimated the population density for Lycalopex spp. (two species together), C. chinga and Leopardus guigna (Table 1), as they are more abundant so that the confidence intervals are acceptable. The population densities of Lycalopex spp., C. chinga and Leopardus guigna at the three survey sites (Alto Colorado (MP), Callihue (SF), and La Estrella (TF)) are 4.1–4.8 individuals/km2, 1.8–2.2 individuals/km2, and 0.7–1.3 individuals/km2, respectively.

The simulated movement of Lycalopex spp. at the three sites, as well as the estimated population density and confidence intervals (95%) are shown in Fig. 4 & Figs. S3–S4 (for Leopardus guigna & C. chinga). We can see Lycalopex spp. are more evenly distributed in Alto Colorado and La Estrella compared to Callihue. C. chinga and Leopardus guigna exhibit spatial heterogeneity across the three survey sites.

Table 1 The estimated population density and 95% confidence intervals at the three sites for Lycalopex spp., C. chinga and L. guigna.

	Population density (Individuals/km2)	
	Lycalopex spp.	Conepatus chinga	Leopardus guigna	
Alto Colorado (MP)	4.8 (2.0, 6.6)	2.2 (0.6, 3.9)	0.7 (−0.3, 1.6)	
Callihue (SF)	4.1 (0.7, 7.6)	2.1 (1.2, 3.2)	1.3 (0.6, 2.1)	
La Estrella (TF)	4.6 (−0.1, 8.8)	1.8 (1.1, 2.9)	1.1 (0.1, 2.4)	

Figure 4 One case of simulated movement of Lycalopex spp. at the three sites (upper panels) and the estimated population density (with probability densities) (lower panels).

Discussion

Species-specific density is an essential parameter for evaluating population stability and ecosystem health both intra and extra guild (Elkinton, 2000). The resource dispersion hypothesis predicts that animal density increases and home range size decreases as resource concentration increases, and may help to explain how carnivores are distributed in environments with a natural gradient (Johnson et al., 2002). Crop farming areas may offer certain carnivore species alternative food sources. For example, small to medium sized carnivores may benefit from rodent populations in agricultural cropping areas (Williams et al., 2018).

In the case of Lycalopex spp., our model shows a high population density in the three landscapes of the coastal mountain range of central Chile, significantly higher than that previously reported by Muñoz Pedreros et al. (2018) for Lycalopex griseus in a similar location (mixed formation of thorny and sclerophyllous forest, and zones with monoculture tree plantations) from central Chile. Our model indicates an even higher density for this genus in the monoculture tree plantations ecotype compared to native forest formations, consistent with the fact that population density has a multivariable dependency, including habitat type and the availability of subsistence resources, primarily the availability of prey that supports the life cycle of carnivore species (Benton, 2009; Duncan et al., 2015; Elkinton, 2000; Tallents et al., 2012). Thus, the vegetation of monoculture tree plantation has been shown to harbour abundant prey populations for mesocarnivores, including introduced exotic species: lagomorphs (Lepus europeus and Oryctolagus cuniculus) and rodents (Rattus norvergicus and Rattus Rattus); as well as native small marsupials and rodents: (i.e., Tylamys elegans, Octodon lunatus, Abrocoma bennetti, Phyllotis sp. Abrothrix sp. Oligoryzomys longicaudatus) (Muñoz Pedreros et al., 2018; Ramírez-Alvarez, 2018). This prey availability and trophic dynamics with carnivores in the monoculture tree plantations ecotype would have a temporal limitation and seasonal adjustments directly related to the planting-harvest cycle of these monocultures (Villaseñor, Escobar & Estades, 2013; Pliscoff et al., 2020). In this trophic dynamic of monoculture plantations, we must also consider that, according to Coon et al. (2020), prey vulnerability (greater exposure of the prey that allows for easier capture by the predator) may be more important than prey availability, following what Hopcraft, Sinclair & Packer (2005) called the ‘ambush-habitat hypothesis’ which proposes that predators choose habitats not based on prey density or encounter rates, but on other factors such as prey vulnerability and minimization of bodily risk.

According to the IUCN (2025), native rodents distributed in our study area show stable population trends at a global level, with the exception of Abrothrix longipilis and Octodon lunatus, which show a decreasing trend. However, there are no recent studies focused on these species in central Chile. In contrast, there has been a clear increase in the abundance and availability of lagomorphs that have gradually colonized these territories. These lagomorphs display high reproductive efficiency, typical of highly adaptable pest species (Camus, Castro & Jaksic, 2008; Bonino, Cossíos & Menegheti, 2010).

Several recent dietary studies have shown that lagomorphs and exotic rodents are common prey for native Chilean carnivores and birds of prey (e.g., Escobar & Jackson, 2023; Castillo-Ravanal, Vallejos-Garrido & Rodríguez-Serrano, 2021; Pizarro et al., 2021). Given this, it is reasonable to question the impact of this introduced prey on predator population dynamics: if this supplemental food source were not available, how would it affect the population dynamics of these predatory species?

Buenavista & Palomares (2018), found that the frequency of exotic mammals in the diet of South American carnivores was at an average of 21%. The most common prey-carnivore interactions involved exotic lagomorphs. Their analysis reviewed dietary studies of native carnivore species also found in central Chile, including Lycalopex culpaeus, Galictis cuja, and Conepatus chinga. Their conclusions align with ours, indicating that exotic mammals in South America can establish new food web interactions within the native carnivore community and serve as a significant food resource, especially in human-modified landscapes where native prey populations have declined. This highlights the role of exotic mammals in supporting the conservation of native carnivore populations.

Barbar, Hiraldo & Lambertucci (2016) conducted a meta-analysis of dietary studies involving 43 predator species that consume lagomorphs as exotic prey in South America and Oceania. They found an average lagomorph-predator link of 20%, indicating a strong interaction. For comparison, the average link with native prey (excluding lagomorphs) is around 24%, which decreases to 17% when lagomorphs are present in the predators’ diets. Their conclusions suggest that when exotic species such as lagomorphs are introduced into a new environment, they can become a critical resource for predators, potentially disrupting the existing food web and creating an unstable balance. Any disruption to this interaction could have catastrophic consequences for native biodiversity, either by directly affecting predator populations or indirectly impacting native prey through competition.

For our study, regarding introduced exotic lagomorphs and rodents, we infer that an ecological subsidy dynamic (Gompper & Vanak, 2008; Newsome et al., 2014) of non-native diet has developed over the years, favoring native carnivores (and birds of prey), which have gradually come to recognize these species as available prey items, focusing on their consumption given their greater biomass and optimization of energy balance, meanwhile native prey items gradually and inversely decrease, because of habitat loss (Ramírez & Simonetti, 2011).

In all the landscapes studied here, we detected a total of 34 photographs of Lycalopex spp. making lagomorphs as prey items (with no significant differences in the number of photos per landscape), providing more evidence that, in the current wild-rural landscapes of central Chile, the population stability of native carnivores and birds of prey is likely determined by the presence and availability of introduced exotic prey (Muñoz Pedreros et al., 2018; Vallejos-Garrido et al., 2024). Therefore, the population control of these latter species (as they are considered pests) will directly impact the population trends of native carnivores and birds of prey, a situation that should be considered by the authorities in their pest control management designs and hunting legislation. For example, prohibiting the use of snare traps (called “huaches” in Chile) in these wild areas, since they do not discriminate between exotic and native species, would help maintain a natural trophic balance with less impact on local mesomammals. Lycalopex genus has the highest population density compared to the other mesopredators in the three studied landscapes. This finding, confirms this genus of native carnivores, as the habitat and diet generalists, are most abundant in the Coastal Range of central Chile (Zuñiga, Muñoz Pedreros & Fierro, 2008; Zuñiga, Muñoz Pedreros & Fierro, 2009; Zúñiga et al., 2022; Ramírez-Álvarez et al., 2023).

For our study, it is essential to consider the ongoing debate regarding hybridization within the Lycalopex genus, which includes several species of South American foxes. This debate has significant implications for species classification and conservation. Hybridization may be occurring between the pampas fox (Lycalopex gymnocercus) and the Andean fox (Lycalopex culpaeus). Mitochondrial DNA studies have revealed instances where individuals classified as Lycalopex gymnocercus exhibited haplotypes associated with Lycalopex griseus, suggesting potential hybridization or introgression events. This finding raises questions about the geographic distribution and reproductive isolation of these species (Tchaicka et al., 2016; Favarini et al., 2022). Another study documented a canid in Brazil that displayed phenotypic traits intermediate between domestic dogs and pampas foxes, indicating interspecific hybridization (Szynwelski et al., 2023). Genetic analyses confirmed this canid as a hybrid, marking the first documented instance of such hybridization between these two species in South America (Szynwelski et al., 2023). The potential for hybridization poses significant concerns for the conservation of wild Lycalopex populations. Hybridization with domestic dogs can lead to genetic dilution and increased vulnerability to diseases transmitted by domestic animals. Therefore, further investigation into the ecological and genetic consequences of such interbreeding is necessary.

Given the practical difficulties in identifying specific species within the Lycalopex genus through visual determination alone—due to potential ambiguity in analysing camera trap images, especially at night, and taking into account the possibility of hybridization—our findings should be interpreted with caution. Inferences should be limited to patterns observed at the genus level, such as density, diet, or habitat use. This approach prevents speculative conclusions and avoids species-specific interpretations of ecological parameters, behaviors, and environmental interactions derived from our research methods.

The few records of G. cuja in this study (n = 38) were not enough to run our model consistently, and therefore, we were not able to estimate a population density for this species. This low detection rate contrasts with the results of Zuñiga, Muñoz Pedreros & Fierro (2009) in southern Chile, who described that this species has a marked preference for fragmented monoculture tree plantations, even above the use of native forests and grasslands.

For Leopardus guigna, our results represent the first density estimation for the species in central Chile ecotypes, being higher than reported by Sanderson, Sunquist & Iriarte (2002) for fragmented landscapes in Chiloé island, and Dunstone et al. (2002) for the ’Evergreen Patagonian Forest’ ecotype in the Austral zone of the country. Even when considering habitat and interspecific variables, it falls within the expected range for the genus Leopardus (Di Bitetti, Paviolo & De Angelo, 2006; de Oliveira et al., 2008; Pereira et al., 2010). Our study is consistent with the ability of this species to adapt to different landscapes (Acosta-Jamett & Simonetti, 2004; Gálvez et al., 2013; Beltrami et al., 2021), including exotic monocultures and fruit tree plantations (Ramírez-Álvarez et al., 2023). Some of these habitats may be occupied only as hunting or transit sites between areas of dense native vegetation where permanent refuges and reproductive sites are located. As with most felids, vegetation cover is an important ecological requirement for Leopardus guigna, used for stalking prey and reproduction (Palomares et al., 2000). However, it is not an obligatory requirement for the presence of Leopardus guigna, as shown by several photographs of this study, where the species appears to be moving in the total absence of vegetation cover in Monoculture plantations and Thorny forest. Our study reveals a lower density of Leopardus guigna in forest monoculture plantations, which may be attributed to the increased human activity inherent in this type of landscape (including patrols, firebreaks, planting-harvesting operations, and domestic dogs). This heightened human presence likely shapes the distribution of this cryptic species, which is highly averse to human interaction.

For C. chinga, our study also constitutes the first report of population density for central Chile, being consistent with densities reported for the species in other ecotypes of South America (Cofré & Marquet, 1999; Castillo et al., 2011; Kasper et al., 2012). In line with what was reported by Vallejos-Garrido et al. (2024), this species is one of the least studied native carnivores in central Chile, and the lack of knowledge about its natural history negatively contributes to its future conservation. In the context of this same study, a sequence of interspecific agonistic interaction between this species and Lycalopex griseus has been recorded and reported (Ramírez-Alvarez & Kaiser, 2024).

The presence of these different species occupying the same landscapes, suggests that both the homogeneity and heterogeneity of these landscapes facilitate the coexistence of these native carnivores (Pereira et al., 2012), and that the subsistence resources in these areas are sufficient to avoid selection by habitat, but not by niche or by daily activity (Benson & Chamberlain, 2007; Fernández, Delibes & Palomares, 2007).

To estimate animal density, we used animal movement parameters from Chinese animals. Although the size and behavior of the eight selected animals are similar to those in our study areas, some uncertainties may persist. Such uncertainty may lead to biased density estimation, and it is unclear whether the population density is being overestimated or underestimated. There is a clear need for further research to refine our models and reduce uncertainties in predicting animal behavior and ecology across different regions.

There is an urgent need to develop conservation strategies at the landscape level for these vital communities of native carnivores that play a crucial role in sustaining a healthy ecosystem in central Chile. Preserving native vegetation cover is essential for sustaining viable and abundant populations of native carnivores, as has been stated in previous studies in central Chile Mediterranean landscapes (Garcia et al., 2020; Beltrami et al., 2021; Ramírez-Álvarez et al., 2023). Conservation strategies in these landscapes should also protect vegetation corridors crucial for dispersal among subpopulations to secure the supply of immigrants and thus contribute to metapopulation persistence (Acosta-Jamett et al., 2003), especially in the case of subpopulations affected by fragmentation or human impacts (Sweanor, Logan & Hornocker, 2000; Pliscoff et al., 2020).

Habitat fragmentation and domestic dogs are threatening terrestrial carnivores, the spatial use of domestic dogs increases with habitat destruction, and domestic dogs and habitat destruction drive the spatial use of native carnivores in degraded agricultural landscapes (Malhotra, Jimenez & Harris, 2021). Therefore, it is essential that future studies investigate the effect of these threats on the density of our native carnivores. The implementation of human-carnivore coexistence formulas, which allow the sustainable balance between habitability and human development, and the conservation of healthy native carnivore populations is one of the most difficult challenges in the currently increasingly anthropogenic world and global change scenario (Chapron & López-Bao, 2016; Lamb et al., 2020).

The delicate balance between the consumption of exotic prey versus native prey in the diets of carnivores in central Chile warrants ongoing research. This is particularly important given the critical role of these animals in maintaining ecological stability and providing ecosystem services. Special attention should be paid to the effects of functional or ecological extinctions—situations where a species’ abundance declines so drastically that it can no longer fulfill its ecological functions—as well as the extinction of ecological interactions, such as ecosystem services, which often precede local species extinctions (Figueiredo et al., 2019; Ramírez-Alvarez et al., 2025).

Conclusions

This study estimated the population density of native carnivores in central Chile using camera trapping data and a simulated movement model. The main conclusions are: Lycalopex spp. (Lycalopex griseus and Lycalopex culpaeus) had the highest population density, ranging from 4.1 to 4.8 individuals/km2. This is the first report of population density estimates for Conepatus chinga (1.8–2.2 individuals/km2) and Leopardus guigna (0.7–1.3 individuals/km2) in landscapes of central Chile. The results highlight their habitat preferences and resource availability across landscapes. The study underscores the complex interactions between carnivores and their environment, emphasizing the role of introduced species in shaping native carnivore populations. This provides insights into the adaptability of carnivores and its implications for biodiversity conservation in central Chile. This research offers novel population density estimates for native carnivores and suggests that introduced prey may subsidize the population dynamics of some species, particularly Lycalopex spp., in human-modified landscapes of central Chile. The findings contribute to a better understanding of carnivore ecology and conservation in this biodiversity hotspot.

Supplemental Information

Supplemental Information 1 The key movement parameters (home range size, movement segment length, standard deviation of angular deflection) of 14 species calculated based on footprint chain data (Li et al., 2022)

Supplemental Information 2 The results of camera trapping at the three sites for Conepatus chinga. The circle sizes indicate the number of effective pictures

Supplemental Information 3 The results of camera trapping at the three sites for Leopardus guigna. The circle sizes indicate the number of effective pictures

Supplemental Information 4 One case of simulated movement of C. chinga at the three sites (upper panels) and the estimated population density (with probability densities) (lower panels)

Supplemental Information 5 One case of simulated movement of L. guigna at the three sites (upper panels) and the estimated population density (with probability densities) (lower panels)

Supplemental Information 6 Camera trapping data used for estimating density of Chilean carnivores

We thank the Chilean Agriculture and Livestock Service (SAG) for institutional support. To Simon Cox and Ivan Salgado for his valuable help on field and data review.

Additional Information and Declarations

Competing Interests

Author Contributions

Data Availability

The authors declare there are no competing interests.

Diego Ramírez-Alvarez conceived and designed the experiments, performed the experiments, prepared figures and/or tables, authored or reviewed drafts of the article, and approved the final draft.

Xinhai Li conceived and designed the experiments, analyzed the data, prepared figures and/or tables, authored or reviewed drafts of the article, and approved the final draft.

The following information was supplied regarding data availability:

Raw data is available in the Supplemental Files.

Code is available at GitHub and Zenodo:

https://github.com/Xinhai-Li/cameratrapR.

Xinhai Li. (2024). Xinhai-Li/cameratrapR: cameratrapR (v1.0). Zenodo. https://doi.org/10.5281/zenodo.14335558.

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
