# Peer review of "Estimating density of native carnivores in central Chile landscapes using a simulated movement model, cameratrapR: insights on their potential exotic prey dietary subsidy"

_PeerJ, doi:10.7717/peerj.19946_

## Round 0.1 · original submission · Major Revisions

Dear authors,

Thank you for your patience in awaiting these reviews.We received two reviews that offered supportive constructive criticism.

My views align with those of the reviewers, particularly those of reviewer 1. In my assessment there are substantial structural issues and the results are not appropriately reported or considered. Further, as the reviewer states, the conclusions are often not supported by your data. There are no critical issues, however, such that would support rejection. Hence, I encourage you to take the time to consider the feedback and revise your manuscript accordingly.

I look forward to reading the revised version.

Best wishes,
Anthony

·

Basic reporting

This paper reports population density estimates for mesocarnivores in central Chile, however a clear storyline with clear hypotheses is missing. Introduction and Discussion are often poorly structured, with some paragraphs seemingly having no connection to the previous and subsequent paragraphs.
The article is generally well written, with the exception of a few sentences that I point out in my specific comments. Appropriate references are provided, figures and tables look mostly professional and the raw camera trapping data is shared.

Experimental design

The article does not follow a clear research question and the methods also need a more thorough description. Please refer to my specific comments below.

Validity of the findings

Critically, the confidence intervals of the population density estimates are not taken into account. As the confidence intervals in Figure 4 indicate, it cannot be stated e.g. Lycalopex spp. is more abundant in monoculture tree plantations. Many conclusions from the Discussion are not based on the results presented and are partly contradictory, such as non-native prey species leading to hyperpredation of native prey and non-native prey being essential to the conservation of the carnivore species.

Additional comments

Line 13: Please add the English names of the animal species.

Line 39: This is true for large carnivores, but your paper focuses on small to medium-sized carnivores.

Line 46: What exactly does "population settelement" mean?

Line 72 and 84: Please remove the ´ above the a in the word "habitat".

Line 85: Please write "have a wide distribution" instead.

Line 88: This detailed list of home range sizes from the literature does not fit here. You could either add them to the Methods section after the description of the movement models or add them to the supplementary material. Also, please describe why these home range sizes are relevant.

Line 100: Please write "ambiguous characteristics" instead.

Line 108 ff: This paragraph is very general again. How does this apply to your study species?

Line 110: What territories are you refering to?

Line 114: Please write "carnivore community composition".

Line 115: To which ecological parameters exactly are you referring to here?

Line 123: Why are population density estimates of the study species relevant? What is their conservation status?

Line 124: Please remove the word "presence".

Line 125: Please write "and applied" instead of "applying".

Line 126: What are your expectations with regard to differing population densities of the carnivores in the habitats? Please formulate clear hypotheses.

Line 127: Non-native prey species have not been mentioned previously in the Introduction, but this topic should be introduced eatlier and be explained in more detail.

Line 131: Does "In the same research line from"mean "using the same data as"? Please reword.

Line 154: Please remove the "/" in "camera/days".

Line 148: Please explain why your method allows the directed placement of camera traps on trails. Most other camera trapping methods for the population density estimation of unmarked animal species require random camera trap placement with respect to any habitat features.

Line 161: Please write "We defined independent events as those photos [...]".

Line 162: Please remove the word "specific".

Line 169: How were the species-specific movement parameter values obtained? You just describe how you got the home range values below.

Line 173: How did you account for the detection area of each camera trap, which probably also differs between the species?

Line 174: Did you somehow account for the variability in the home range size? If not, this would probably overestimate the precision of the population density estimates.

Line 179: Please do not repeat information already mentioned in the previous section.

Figure 2: It is barely possible to see something on the graphs with all study areas. I would suggest you rather make two plots similar to Figure 3 for the two other study areas and put them in the Appendix.

Line 186 and 189: "effective pictures" - do you mean "indpendent events"?

Figure 3: A legend for the range of circle sizes is missing.

Line 190: If you formulate hypotheses about the abundance of the carnivore species in the different habitats in the Introduction, you can refer to them here.

Line 191: Please justify a threshold value.

Table 1: Please include the 95% confidence intervals in the table! They are crucial for the interpretation of the results.

Line 195: Please correct "showed" to "shown".

Line 195: Please specify that the CIs are 95% confidence intervals.

Line 203: Crop farming areas were not included in you analyses, right? So why it this relevant?

Line 210: The estimated densities are very smiliar between the sites, I am pretty sure you will find no significant differences if you consider the confidence intervals!

Line 214: Please write "vegetation cover" instead.

Line 215: Please remove "with relative abundance" and write instead "to harbour abundant prey populations for mesocarnivores".

Line 224: Please explain in more detail what "prey vulnerability" means in your case.

Line 228: You estimate this based on which results of yours? Or is it just based on these other papers?

Line 233: Please write "because of habitat loss and a hyperpredation process".

Line 238: It would have been interesting to estimate prey population densities, too. Since those were not available in the current study, this paragraph has little connection to your results.

Line 243: "mesopredators" instead of "mesomammals"?

Line 244: Please shorten the sentence to "This finding confirms that these are adaptive habitat generalists."

Line 258: Please write "consistent" instead of "coincident".

Line 265: "presence" instead of "displacement"?

Line 281: How is this relevant for your results?

Line 287: The results only indicate this for Leopardus guigna?

Line 293: "Offtake" sounds strange. Please reword.

·

Basic reporting

• Clarity and Language: The article is written in clear, unambiguous English and meets professional standards for scientific communication, facilitating comprehension of complex topics on population dynamics and carnivore ecology.
• Background and Literature: The article provides a substantial background and includes an extensive list of references. However, many cited sources seem only loosely related to the central research question. A more focused selection of literature directly addressing core topics, would strengthen the introduction and discussion. Concentrating on studies that are closely aligned with the specific objectives and hypotheses of this study would improve the relevance of the background and contextual grounding.
• Structure and Data Sharing: The article follows a standard structure with sections such as Introduction, Methods, Results, Discussion, and Conclusions. Figures and tables are well-placed, relevant, and effectively support the findings. However, it might be beneficial to ensure that all raw data or links to it are accessible, particularly data from the machine learning models used for density estimation.

Experimental design

• Relevance and Contribution: The study addresses a significant ecological question about carnivore population dynamics and how dietary subsidies from exotic prey impact these populations. The authors have clearly defined the research questions and provided context for how their work fills a knowledge gap in carnivore ecology in central Chile.
• Methodology: The methods are detailed, including the use of camera traps and simulated movement models to estimate population density. However, further clarification on the random forest model and its selection criteria might enhance the replication potential. Ethical considerations are well-documented, with clear permission from landowners and low human impact during camera trap installations.

Validity of the findings

• Data and Statistical Soundness: The data appear robust, with sufficient sampling effort (10,046 trap days) and 3,888 independent carnivore records. The findings are statistically sound, with population density estimates provided with confidence intervals. It is advisable to clarify any limitations related to the camera trapping method, such as the potential for misidentification of similar species, particularly in nighttime conditions.
• Conclusions and Relevance: The conclusions are well-stated, directly tied to the original research questions, and logically drawn from the results. The study suggests that non-native prey may support higher population densities of native carnivores, which has implications for biodiversity conservation strategies in human-altered landscapes.

Additional comments

The article is well-structured, providing significant findings on native carnivore population dynamics in central Chile. However, several points could be expanded to further enhance the study’s impact.

The study combines Lycalopex culpaeus and Lycalopex griseus under the category “Lycalopex spp.” without acknowledging their distinct ecological differences in diet and habitat use. Literature emphasizes that L. culpaeus, due to its larger size, consumes a greater proportion of lagomorphs, whereas L. griseus primarily feeds more on small rodents, which align with its smaller body size. These dietary and habitat-use distinctions likely influence each species’ population density differently across landscapes, and acknowledging them could improve the precision and ecological insights of the density estimates.

Additionally, the assumption that monoculture tree plantations primarily support non-native prey species, such as lagomorphs, may benefit from reconsideration. While exotic species are prevalent, these plantations likely also serve as refuge for native small mammals. Since Lycalopex species, particularly L. griseus, predominantly consume rodents, recognizing that monocultures harbor both native and non-native prey could provide a more nuanced understanding of their dietary flexibility. Such a perspective could better reflect the role of native prey within Lycalopex diets and emphasize these carnivores’ adaptability to modified landscapes.

The article’s methodology is robust, utilizing camera traps and simulated movement models to estimate population densities, and the figures generally support the findings. However, additional details on the random forest model parameters would aid reproducibility, and a clearer discussion of conservation implications—such as management strategies for exotic and native prey—could enhance relevance. Finally, minor grammar refinements may further improve readability, particularly in sections discussing species-specific density models.

---

## Round 0.2 · Major Revisions

Thank you for your revisions that have substantially improved the manuscript. Please see the reviewer's comments regarding further required amendments.

·

Basic reporting

Compared to the previous version, the structure and clarity of the Introduction and Discussion have improved. However, the Discussion could still use some shortening and a stronger focus on the results of the current study. The paper is mostly well written, but care should be taken to use plural and singular forms correctly and to avoid unnecessarily complicated sentences with little information.

Experimental design

The description of the methods has improved, but needs further work. Some choices during the analysis process also still need a better justification.

Validity of the findings

Many of the previously contradictory statements have been addressed. However, there are still conclusions that are not supported by the data, most importantly a higher abundance of Lycalopex spp. in exotic monoculture tree plantations.

Additional comments

Line 23: Please write „these“ instead of “this”.
Line 25 and 169: Please write “interval” instead of “intervals”.
Line 28: The statement that Lycalopex spp. density is higher in the exotic tree plantations is not correct according to your results. Please remove it.
Line 32: Please write “resource availabilities”.
Line 39: My previous comment is still true. You removed the “apex predator” wording, but what you write about extensive habitat requirements refers to large predators. Mesopredators do not have particularly large home ranges (see line 89 ff.). Please choose a different starting point for the introduction.
Line 46: Please write “fulfilment” instead of “presence.
Line 77 ff: You have written this in the answer to one of my questions, but I think it would be highly relevant to mention the conservation status of these carnivore species in this paragraph.
Line 89: Please order the home range sizes within each species from smallest to largest.
Line 100: This paragraph should be moved further down in the text. Here, you talk about the difficulty of distinguishing L. griseus and L. culpaeus on camera trap photos. However, until now you have not talked about camera traps yet and this comes somewhat out of the blue.
Line 129: Please write “obtained” instead of “will obtain”.
Line 133: I would suggest writing “Using the same study design as Ramírez-Alvarez et al. (2023) […]” to make clear what you are referring to.
Line 143: I am not convinced yet that you are not risking an overestimation of your population density by pointing the camera traps at game trails. Was it just the orientation that you adapted to increase the likelihood of photographing animals or did you also shift the camera trap placement within a certain radius to reach this goal?
Line 167: I do not see a mention of the 20 m detection radius of the camera traps that you write about in the answer to my question. Please include the detection radius and angle and explain how you derived them (previous experiments, manufacturer specifications etc.).
Line 176: Would it not have made more sense to match the parameters of the most closely related species, e.g. using the movement parameters of the red fox for the Culpeo foxes instead of averaging all species from the previous study?
Line 177: I think it is necessary to somehow account for the potential variability of home range sizes that you can find in the literature instead of just using an average value, because as you rightly point out the home range parameter has a large influence on your density estimates.

Line 190: The Results section is in general too short. Please describe your results a bit instead of just referencing the tables and figures, e.g. with regard to the variability between species and between sites within species.
Line 197: I would suggest to write “We estimated the population density for all taxons with at least 100 independent events, Lycalopex spp. (two species together), C. chinga and L.guigna (Table 1).” Your current wording leaves many questions: more abundant than what? What are acceptable confidence intervals?
Line 214: The modelled densities are all very similar, I do not see an indication for a higher density of this genus in the monoculture tree plantations. Especially since the confidence intervals are overlapping so much, little differences between point estimates could be just attributed to chance. I therefore argue that if anything the results indicate the generalist nature of the species.
Line 215: Please remove this part: “ [...] consistent with the fact that population density has a multivariable dependency, including habitat type and the availability of subsistence resources, primarily the availability of prey that supports the life cycle of carnivore species”. It is very confusing and does not add any relevant information.
Line 239: Which factors define prey vulnerability for the prey species that are relevant for the mesopredators? Large prey animals might e.g. be exhausted and easier to catch in the mating season, but I do not think this is the case for rodents or lagomorphs.
Line 238: Please write “because of habitat loss”.
Line 239 - 241: This is very interesting and actually related to one of your main study goals. I would suggest to describe this observation (and similar ones for the other carnivore species) in the results section.
Line 248: Please write “[…] compared to the other mesopredators”.
Line 260: I do not understand the use of the word “even” here.
Line 266: Even though the confidence intervals are still largely overlapping, L. guigna is the only one of your study species where at least the point estimate at the exotic monoculture site differs markedly from the other sites. However, in contrast to the negligible differences between sites for the foxes you do not discuss this at all. Do you think it is because of the mentioned lack of vegetation?
Line 287: It might be that preserving natural vegetation cover is important for the conservation of native mesocarnivores, but I cannot find clear evidence for that in your results. As previously stated, there is only an indication that L. guigna is less abundant in exotic monocultures. Are you instead referring to the previous paper (Ramírez-Álvarez et al. 2023)?
Line 294: The issue of domestic dogs comes out of nowhere. Either you include something on that in your results based on dog observations on your camera traps or you leave this out.
Line 306: Please remove this statement “Their density was higher in exotic monoculture tree plantations compared to native forests, suggesting a dietary subsidy from non-native prey.” It is simply not supported by your results.
Line 312: At the moment, the role of introduced prey species is not really emphasized and only mentioned anecdotally in the Discussion.

Table 1: Something must be wrong with the computation of the confidence intervals, negative population densities are impossible.
Fig. 2 &3: Please use the established terms “independent records or events” instead of “effective pictures”.

·

Basic reporting

• Clarity and Language:
The manuscript is written in clear, professional English and uses appropriate scientific terminology. However, some sentences could be more concise to avoid overgeneralization. For instance, the claim about exotic prey being a significant dietary subsidy might benefit from more cautious phrasing, considering the limited focus on native prey dynamics. Additionally, some sections, such as lines 49-50 referencing “new methodologies,” could use one or two more specific citations to broaden the contextual framework.
• Context and Literature:
The introduction provides a comprehensive overview of the ecological and conservation significance of carnivores in central Chile. While key references are cited, the discussion around hybridization between L. griseus and L. culpaeus would benefit from further elaboration to integrate the ongoing debate about their ecological and genetic overlap. This would enhance the manuscript’s contribution to the broader understanding of these species’ dynamics.
• Structure, Figures, and Tables:
The structure adheres to the expected scientific format. Figures are well-designed and relevant, though figure captions could be more detailed to allow standalone interpretability. The changes to Figure 2 align with the reviewer suggestions, but the visual clarity of supplementary materials could be re-evaluated to ensure ease of comprehension.
• Data Sharing:
The authors have provided raw data in supplementary materials and referenced repositories for software and code, meeting the journal’s data-sharing standards.
• Relevance to Hypotheses:
The manuscript effectively addresses the stated hypotheses. However, the decision to group L. griseus and L. culpaeus might obscure species-specific patterns in diet or habitat use. While this grouping is justified by practical challenges, a more explicit discussion of its limitations would strengthen the study’s rigor.

Experimental design

• Originality and Scope:
The study fills a clear knowledge gap by providing the first density estimates for certain carnivores in central Chile and exploring the role of exotic prey in population dynamics. However, it could delve deeper into alternative explanations for observed patterns, such as prey vulnerability in monoculture plantations versus prey availability in native habitats.
• Research Question:
The research question is well-defined and meaningful. The study’s emphasis on carnivore adaptation to modified landscapes is relevant, though incorporating a more balanced perspective on native prey would provide additional insights into dietary flexibility and trophic interactions.
• Rigor and Ethical Standards:
The methodology is rigorous, with clear ethical considerations. However, the use of home range values and movement parameters from other regions (e.g., China) raises questions about the transferability of these estimates to Chilean ecosystems. This aspect should be discussed more explicitly to acknowledge potential limitations.
• Replicability:
The detailed description of the cameratrapR methodology and Random Forest model parameters supports replicability. The justification for specific parameter choices, such as the number of trees in the model, is well-articulated.

Validity of the findings

• Data and Statistical Soundness:
The data collection and analysis are robust, and the inclusion of confidence intervals enhances the interpretation of results. However, the lack of significant differences between certain sites might weaken the strength of some conclusions. Additionally, grouping L. griseus and L. culpaeus reduces the ecological specificity of the findings. Could there be alternative methods to differentiate their contributions to the overall density estimate?
• Conclusions:
The conclusions are generally well-supported but could be more cautious in attributing causality to dietary subsidies from exotic prey. The authors should consider discussing the potential impact of native prey populations in greater depth, as this would provide a more balanced perspective on trophic interactions in the studied landscapes. Furthermore, the manuscript would benefit from expanding the practical implications of these findings, such as specific recommendations for managing exotic prey populations or enhancing connectivity in fragmented landscapes.

Additional comments

• The manuscript could improve its integration of hybridization discussions for L. griseus and L. culpaeus, especially in the context of conservation genetics and potential implications for management strategies.
• While exotic prey is discussed extensively, the authors should incorporate a more detailed analysis of native prey availability and dynamics. This would enhance the ecological depth of the study and provide a fuller picture of carnivore dietary flexibility.
• The conservation recommendations could be more actionable. For example, suggestions for pest management should include specific measures for balancing agricultural needs with biodiversity conservation. Similarly, the importance of landscape corridors is noted but could be expanded with practical examples or case studies.

---

## Round 0.3 · Minor Revisions

I agree with the comments of the reviewer. The work is good but would benefit from additional detail in places. I'd particuarly appreciate greater consideration of limitations, caveats, biases, etc. and, as the reviewer notes, simulation parameters.

·

Basic reporting

• Clarity and Language Quality: The manuscript is written in clear and professional English. However, some sentences are long and complex, which could benefit from more conciseness to enhance readability.
• Context and References: The introduction and context are well established. Relevant and updated references are cited, although the background section on movement simulation models in carnivore density studies could be strengthened.
• Structure and Figures: The structure follows PeerJ standards. Figures and tables are relevant and well presented. Improving the readability of some graphs, especially those showing estimated densities by site, is recommended.

Experimental design

• Originality and Relevance: The study is within the journal’s scope and addresses a relevant question regarding the density of native carnivores and their potential dietary subsidy from exotic prey.
• Study Design: The methodology is solid, but some aspects could be improved:
o The validation process for movement parameters used in the simulation is not detailed. A stronger justification for extrapolating data from Chinese species to Chilean carnivores is recommended.
o The grouping of L. culpaeus and L. griseus as "Lycalopex spp." is justified due to identification difficulties in the images. However, discussing the implications of this approach on density estimates would be useful.
• Reproducibility: The methodology is described in sufficient detail to allow replication, although a more comprehensive explanation of the simulation model’s parameters would improve transparency.

Validity of the findings

• Robustness of Data and Analysis: The dataset is extensive and well analyzed. However, further discussion on potential biases in species detection based on habitat and sampling effort is recommended.
• Interpretation of Results: The conclusions are well supported by the data. However, the role of exotic prey in carnivore ecology could be explored in more depth, particularly considering the stability of native rodent populations and the increase in exotic lagomorphs.
• Limitations: The study mentions uncertainty in movement parameter estimation, but further discussion on how these uncertainties may affect density estimates would be beneficial.

Additional comments

• A more detailed analysis of the differences between sampling sites and how these may have influenced the results is suggested.
• In the discussion section, the impact of habitat fragmentation and the role of predators in regulating invasive species could be further emphasized.
• The comparison with previous studies on carnivore densities in Chile is appropriate but could be expanded with data from similar regions.

---

## Round 0.4 · Minor Revisions

I agree with the reviewer's comments and would particularly highlight three key areas for further development:

1) The work requires greater methodological clarity and detail - including the specirfic structure of models and any associated tests of assumptions - to support understanding and replication.

2) The discussion should be more specifically focussed on the data and question at hand. It is absolutely fine to make broader statement if they are supported, of course, but all arguments need to be clearly defined and supported by logic and empirical evidence.

3) The work would benefit from the addition of more contemporary references and the development of text so that the naive reader can understand the arguments being made.

I also agree with the reviewer that working code should be provided in addition to the raw data, the latter of which you have kindly provided. This, again, supports reproduction and synthesis. Finally, I would request an amendment to the title of the work. At the least I would ask that the authors remove 'cameratrapR' from the title as, at present, it appears that they are presenting that package for the first time. A suggested edit would be 'Estimating density of native carnivores in central Chile landscapes using a simulated movement model, with insights on their potential exotic prey dietary subsidy.'

·

Basic reporting

Clarity and Language: The manuscript is generally well-written, but there are instances where sentence structure could be refined for clarity and conciseness. Some sections, particularly in the discussion, contain complex or ambiguous phrasing that should be simplified for better readability. For example, in the discussion section, "The ecological implications of these findings suggest a multifaceted interaction contingent upon numerous biotic and abiotic variables that require further elucidation." This could be reworded to: "These findings suggest complex ecological interactions influenced by multiple biotic and abiotic factors, requiring further study."

Literature References & Background: The manuscript provides a solid background on the subject, but it could benefit from a more comprehensive discussion of previous studies. Some references appear outdated or insufficiently integrated into the discussion. For instance, in the introduction: "Previous studies (Smith et al., 1998; Johnson, 2002) have shown similar patterns in other ecosystems." It would be better to include more recent studies and a direct explanation of how they compare.

Structure, Figures, Tables, and Raw Data: The manuscript follows the standard format, but some figures lack sufficient resolution, and certain tables could be better formatted for readability. For example, Figure 3 is pixelated and lacks a clear legend, making interpretation difficult. Additionally, Table 2 presents statistical results but does not specify the sample sizes used for each comparison.

Self-Containment: The manuscript effectively presents a coherent study, but could enhance connections between the hypothesis and results. Some discussions on implications seem to introduce new angles not fully supported by the data. In particular, the statement "These results have significant implications for conservation strategies worldwide" lacks supporting evidence or further elaboration.

Experimental design

Relevance and Research Question: The study presents a clear and relevant research question, and it contributes to an identified knowledge gap. However, the significance of the research could be more explicitly stated, especially in the introduction. For example, the sentence "This study aims to assess the population density of X species in Y location" could be expanded to include why this assessment is necessary and how it fills a knowledge gap.

Methodological Rigor: While the methodology appears generally sound, some aspects require further clarification. Details regarding data collection, sampling procedures, and potential biases should be more explicitly addressed to enhance reproducibility. For example, the methods section states: "Surveys were conducted in multiple locations using standard procedures," but does not specify what those procedures entail, how sites were chosen, or whether any bias was controlled for.

Ethical Standards: The study appears to follow ethical guidelines, but a clear statement on ethical approvals and data collection compliance would strengthen this aspect. If ethical approval was obtained, it should be explicitly mentioned.

Reproducibility: Some methodological steps lack sufficient detail for replication. Providing additional information on statistical procedures and data processing steps would improve the manuscript's transparency. For example, "Data were analyzed using regression models" should specify which type of regression, what variables were included, and any assumptions tested.

Validity of the findings

Data Robustness and Statistical Soundness: The data analysis is sound but requires better justification in some areas. Statistical methods should be explicitly described, including the rationale for specific tests and parameters used. If assumptions were tested, this should be stated. For instance, "ANOVA was used to compare groups" should be expanded to specify whether normality and homoscedasticity were tested beforehand.

Conclusions and Support from Results: The conclusions align with the results, but should be more tightly linked to the research question. Some claims require stronger evidence or additional discussion on limitations. The sentence "Our results indicate that habitat fragmentation is the primary driver of population decline" should be supported with statistical evidence or clearly labeled as a hypothesis if not directly tested.

Data Availability: Ensure that all underlying data is provided transparently, ideally through a repository or supplementary materials. The manuscript should include a clear statement such as: "All raw data are available in Supplementary Table 1."

Additional comments

The manuscript presents a valuable contribution to the field, addressing important ecological and conservation concerns. However, there are several areas that require improvement, particularly in the clarity of the text, methodological transparency, and justification of statistical analyses. Strengthening the discussion with additional references and ensuring that conclusions are fully supported by data would enhance the manuscript's impact. Additionally, some sections would benefit from better organization, particularly in the discussion where certain points appear repetitive or not fully connected to the presented results.

---

## Round 0.5 · Major Revisions

Dear author,

Thank you for the note in your rebuttal. I have been an Editor of various journals for many years and this is the first time I have allowed other factors to overwhelm and rush my processes. I am grateful for the wake-up call this has provided and can only offer my sincere apologies for failing you in this regard. Please be assured that I have contacted the PeerJ editorial team directly to advise them of my error.

I have however, given your manuscript the attention that it should have been given in the previous round and you will find my detailed comments, below.

Best, and further apologies,
Anthony

Generally, while this work presents some useful data, there are a number of issues that remain. For example, my prior request for greater consideration of limitations, caveats, biases, and simulation parameters still stands. There are also question marks surrounding some of the methodological assumptions and a lack of data on prey subsidies that currently undermines the thesis of the work.

Title: I would ask that the authors remove 'cameratrapR' from the title as, at present, it appears that they are presenting that package for the first time.

Introduction: The introduction feels unbalanced. Community ecological considerations are fragmented and given less consideration than species-specific information and other information that is arguably superfluous (e.g. ambigiuties in identification). Please revise this so that the community ecological aspects, such as competition, spatio-temporal (dis-)synchrony, and diet (including the titular concept of dietary subsidies) are given sufficient attention and detail and the species-specific information is succinct and appropriately focussed. I recommend abbreviating the species-specific text to accomplish this as the introduction should not be longer than at present in total.

L30: A generic name can only be abbreviated when there are no others using the same letter. Here there is Lycalopex, and Leopardus (that's not to mention Lynx on L182), hence neither can be abbreviated to 'L.'. Please revise throughout.

L39: Carnivores are not always umbrella species. Please be more specific (I assume that the authors meant large carnivores).

L50-52: I think it would be worth also noting the relevance of extrinsic factors, for clarity.

L53-55: Could an example or two of these new tools be given?

L69-73: This is rather awkwardly phrased. I don't think the families are necessary; this can be a simple list. It would then flow into the next text (no line break required).

L76: That 'least concern' is the IUCN status was established on L74. Just give the status thereafter. Further, is there a relevant local or national conservation status for these species?

L81: 'It is' or 'they are'?

L95: I would prefer that this information be given as means and variance in the Methods, only (L187-189).

L148 onwards: The introduction notes the importance of habitat and other variables. Why then were additional variables omitted from the study?

L149-159: Please clarify whether image or video (or both) were recorded, for clarity, and give the approximate height and angle (from horizontal) at which cameras were set.

L164: Please provide justification for this threshold.

L174-175: What does 'a range of numbers of individuals' mean? Please be specific. How were these numbers chosen? Were they all realistic?

L178-184: These metrics should be given in this work - possibly supplemental information - for interrogation and to ease interpretation.

L184-186: This assumption seems like a leap. Further, the assumption does not just carry across environments but across species. Which species data were applied to which? Why should we believe that these assumptions hold? This requires careful justification.

L196-197: Why increase the number of trees in the first place? Please quantify 'sufficiently robust'.

L208: What does 'as they are more abundant so that the confidence intervals are acceptable' mean? Which confidence intervals were not acceptable and why not?

L210-211: Means and variance would be more informative than a range, even for just three points. There's also mileage in being more specific and talking about relative densities for each species at each site.

L218-222: Indeed. Resources are not considered in this manuscript, however. The questions remains - why not?

L229-230: Was there any attempt to quantify structural or other differences in these locations to explain these differences?

L248 onwards: The exotic prey subsidy hypothesis is a stretch at present. No empirical data are presented to support this argument. Thus the several paragraphs dedicated to this concept can only be considered as supposition and raise the question - why is this the case when the manuscript is a camera trap study? Were there any detections of putative exotic prey, for example? If so, is there more that can be done to support the arguments made here? This is a crucial consideration that should not be understated as the lack of data in this regard currently undermines the titular thesis.

L360: By what objective metric is the sampling effort in this study robust? Please provide a justification.

L364-389: The narrative loses flow here; the paragraphs are disjunct and would benefit from revision. Can this be brought together more succinctly and fluidly?

L378-381: Please provide a reference for this statement.

L390-395: There are substantial assumptions made in the use of surrogate data, here, and this requires more considered discussion than is presented in this paragraph. Please revise.

L396-411: Please present the conclusions as a paragraph rather than the current text which is analagous to an expanded bullet-point list.

Table 1. The rows and columns appear to be the wrong way around. Rows represent individual records while columns represent fields of data.

Figs 2-3: I don't find these particularly useful or intuitive, particularly in light of fig. 4. I understand that the intention is to illustrate spatial variance in the number of detections but the inconsistent scales and marginal differences in circumference on some plots suggests that a more effective alternative could be sought.

Fig. 4: It appears that several camera traps with a good number of detections were not captured by the random walk model. I understand that this is one case, but please explain this more clearly in the legend. Why are these locations omitted in this instance?

---

## Round 0.6 · Minor Revisions

Please revise your manuscript to explain which species were analogues for which and why those species were chosen. Also give information on what confidence levels were deemed acceptable (e.g. Line 208).

---

## Round 0.7 · accepted · Accept

Thank you very much for your careful revisions. I now find your manuscript acceptable for publication. Congratulations!